# Emerging MIMO Technologies for 6G Networks

**DOI:** 10.3390/s23041921

**Published:** 2023-02-08

**Authors:** Victoria Dala Pegorara Souto, Plínio Santini Dester, Michelle Soares Pereira Facina, Daniely Gomes Silva, Felipe Augusto Pereira de Figueiredo, Gustavo Rodrigues de Lima Tejerina, José Cândido Silveira Santos Filho, Juliano Silveira Ferreira, Luciano Leonel Mendes, Richard Demo Souza, Paulo Cardieri

**Affiliations:** 1National Institute of Telecommunications, Santa Rita do Sapucaí 37540-000, Brazil; 2Department of Communications, University of Campinas, Campinas 13083-970, Brazil; 3Centro de Pesquisa e Desenvolvimento em Telecomunicações, Campinas 13083-970, Brazil; 4Department of Electrical and Electronic Engineering, Federal University of Santa Catarina, Florianópolis 88040-900, Brazil

**Keywords:** 6G networks, massive MIMO, XL-MIMO, IRS, cell-free mMIMO

## Abstract

The demand for wireless connectivity has grown exponentially over the last years. By 2030 there should be around 17 billion of mobile-connected devices, with monthly data traffic in the order of thousands of exabytes. Although the Fifth Generation (5G) communications systems present far more features than Fourth Generation (4G) systems, they will not be able to serve this growing demand and the requirements of innovative use cases. Therefore, Sixth Generation (6G) Networks are expected to support such massive connectivity and guarantee an increase in performance and quality of service for all users. To deal with such requirements, several technical issues need to be addressed, including novel multiple-antenna technologies. Then, this survey gives a concise review of the main emerging Multiple-Input Multiple-Output (MIMO) technologies for 6G Networks such as massive MIMO (mMIMO), extremely large MIMO (XL-MIMO), Intelligent Reflecting Surfaces (IRS), and Cell-Free mMIMO (CF-mMIMO). Moreover, we present a discussion on how some of the expected key performance indicators (KPIs) of some novel 6G Network use cases can be met with the development of each MIMO technology.

## 1. Introduction

The First Generation (1G) of mobile communication networks was released in the 1980s when the only functionality was to provide voice communication using analog mobile technology. To address the limitations of analog communication systems, the Second Generation (2G) featured a digital network which became part of every mobile communication system. In the 1990s, data applications such as short message services (SMS) started being supported. The Third Generation (3G) introduced mobile broadband services and enabled new applications such as multimedia message services, video calls, and mobile TV. Improved mobile broadband services, high-definition video streaming, and seamless handover became possible in the Fourth Generation (4G). The adoption of communication based on Internet Protocol (IP) introduced various quality of service (QoS) levels to meet different users’ demands [1].

As a consequence, the constant evolution of wireless technologies led to an exponential increase in connected devices. Thus, the Fifth Generation (5G) of mobile communication networks represents a significant leap to support this demand. With 5G, the network should support stringent requirements in terms of data rate (∼10 Gbps), latency (∼1 ms), connectivity (∼1 million/${\mathrm{km}}^{2}$), reliability (∼$10^{-5}$), and energy consumption [2]. Regarding the expected Sixth Generation (6G), the architecture and performance components remain mostly undefined. However, to meet the anticipated bold key performance indicators (KPIs), many new developments are needed, including on multiple-antenna technologies such as massive Multiple-Input Multiple-Output (mMIMO) [3], extremely large MIMO (XL-MIMO) [4], Intelligent Reflecting Surfaces (IRS) [5,6], and cell-free mMIMO (CF-mMIMO) communications [7].

The mMIMO concept was introduced in [8] and is a vital technology in 5G Networks. In mMIMO, antenna arrays have up to an order of magnitude when compared to conventional MIMO schemes. The deployment of mMIMO schemes can significantly increase the spectral and energy efficiency of communications systems [9]. In addition, we can further increase the number of antennas to deploy huge arrays at the base station (BS), giving rise to XL-MIMO, a promising research direction in multi-antenna technology for 6G Networks [10]. With XL-MIMO, it is possible to increase significantly the spectral and energy efficiency in some potential scenarios of 6G Networks.

Another promising approach for 6G Networks is the CF-mMIMO paradigm, in which the absence of well-defined cell boundaries reduces not only problems inherent to the handover process but also cell-edge user’s limitations and inter-cell interference [11]. Moreover, IRS has been recently proposed and constitutes a new technology that can achieve smart and reconfigurable radio propagation environments. By densely deploying IRS in future 6G Networks while intelligently coordinating the phase and amplitude of their elements, the wireless channels can be intentionally and deterministically controlled to improve the signal quality at the receiver, and consequently, the network capacity and reliability [5,6,12,13,14].

Several recent works provide reviews of the main concepts, advantages, drawbacks, and challenges of one or some of the above-mentioned multiple-antenna technologies [4,15,16,17,18], but they do not focus on 6G Networks architectures or requirements. Moreover, many other works [9,19,20,21,22,23] present a vision of future 6G Networks and describe the concepts of some technologies that are fundamental to meet the anticipated requirements. However, these studies do not discuss the role of emerging MIMO techniques in detail. Notably, in [24] the authors survey multiple-antenna technologies that will likely play key roles in 6G Networks, namely: CF-mMIMO, mMIMO, and IRS. They present the fundamental motivations, key characteristics, and recent technical progress while providing some perspectives for future research directions. However, they do not contextualize the discussion around novel 6G Networks use cases and requirements.

### Contributions

This survey presents a concise review of the main emerging MIMO technologies for 6G Networks, such as mMIMO, XL-MIMO, IRS, and CF-mMIMO communications. Moreover, different from [9,19,20,21,22,23,24], this survey describes in detail some innovative use cases that shall be supported by 6G networks and, from their features, presents an important description of the main requirements to fully implement each use case. In addition, we extend our study by showing how the KPIs of those novel 6G Networks use cases can be met with the development of each MIMO technology. To the best of the authors’ knowledge, no other work in the literature discusses specific technical requirements for 6G use cases and their relation with the MIMO techniques deployment, which is a novelty and a contribution of this manuscript. Moreover, it is important to highlight that the main goal of this paper is not to present a comparison between the MIMO technologies, but to demonstrate the importance of the MIMO technologies to meet the KPIs of innovative use cases of 6G Networks. More specifically, the intention is to present to the readers some vertical sectors that may be supported by the deployment of these emerging technologies. This paper is a result of the 6G Brazilian National Project (Brazil 6G).

The main contributions of this work can be summarized as follows:1.Some future use cases are described with their main stringent requirements that 6G Networks must address.2.The main features, advantages, and related challenges of some emerging MIMO technologies for 6G Networks are presented.3.Some open research topics related to mMIMO, XL-MIMO, IRS, and CF-mMIMO are discussed, giving directions for future works in 6G Networks.

In order to improve the readability of this survey, the organization of the survey is illustrated in Figure 1. More specifically, the rest of this paper is organized as follows. Section 2 presents a review of mMIMO and XL-MIMO. Section 3 discusses the CF-mMIMO approach, while Section 4 reviews the IRS concept. Section 5 gives an overview of some innovative use cases of 6G Networks and presents a discussion about how MIMO technologies are enablers of their stringent KPIs. Section 6 presents a discussion on some open research topics. Finally, Section 7 concludes the paper.

## 2. Massive MIMO and XL-MIMO

The use of large arrays of antennas has been one of the defining characteristics of the 5G Network. Furthermore, known as massive MIMO, this technology was introduced in [8]. By adding elements to antenna arrays of up to an order of magnitude compared to conventional MIMO schemes, mMIMO can significantly increase the spectral and energy efficiency of wireless systems [9]. Thanks to many antennas, it is possible to create highly directional beams and serve multiple users simultaneously using the same time and frequency resources [25]. Furthermore, these highly directional beams make the BSs able to concentrate energy on the intended users [26]. Figure 2 illustrates traditional MIMO, mMIMO, and XL-MIMO systems.

The use of large antenna arrays at the BSs and spatial multiplexing techniques allow the users to share the same frequency bands and, consequently, considerably increase the system’s spectral efficiency. In addition, a large number of antennas at the BS allows a higher degree of freedom, i.e., the transmitted signals at the BS can be designed to add constructively at the users and destructively in other directions. Another advantage of deploying a massive number of antennas at the BS is the capacity to concentrate the energy in a small area, which increases the system’s energy efficiency. To conclude, by using mMIMO systems, it is possible to deal with interference and fading efficiently, and then the system can experience a much-improved performance [27]. Despite the gains obtained through implementing mMIMO in future communication networks, there are numerous open problems with its implementation, as described below.

**Hardware impairments:** mMIMO takes advantage of the law of large numbers to mitigate fading, and, to some extent, interference. However, one significant challenge of mMIMO is the implementation and deployment of massive RF chains and performance degradation due to hardware impairments since low-cost RF chains are adopted to reduce energy consumption and deployment costs [28]. This requires solutions capable of overcoming hardware imperfections such as I/Q imbalance and phase noise.**Mutual coupling and front-back ambiguity:** When modeling antenna arrays, an assumption often made is that the separation between array elements is large enough to keep mutual coupling at negligible levels. However, this is not entirely realistic, especially in the case of many elements deployed as an array of constrained size and aperture elements. Therefore, under such practical conditions, the mutual coupling impacts the system capacity [29,30] and becomes a critical challenge for the mMIMO implementation.**Propagation and channel modeling:** Realistic performance assessments require proper channel characterization and modelling. The behavior of the mMIMO channel, including its correlation properties and the influence of different types of antenna arrays, greatly impacts performance [31,32,33,34].**Precoding:** Interference from multiple users can be mitigated on the transmission side by using beamforming techniques to support multiple data streams. Zero Forcing (ZF) or Minimum Mean Square Error (MMSE) based precoding is simple for a moderate number of antennas. However, the reliance on channel inversions, that is, matrices, can take their complexity and energy consumption to a point difficult to accommodate in massive arrays [25,35,36,37,38].**Detection:** When it comes to data stream separation in conventional systems, maximum likelihood detection is the ideal solution, but its complexity increases exponentially with the number of streams (this makes it challenging to implement in networks supporting mMTC where hundreds to thousands of devices are provided) since estimation and detection are critical issues in mMIMO systems [27].

The previously described challenges of the mMIMO technology lead to open research topics that have considerably attracted the attention of researchers and the industry, resulting in a continuous evolution of the technology. In addition, as previously mentioned, the mMIMO is an important technology for the full development of the 6G Networks and, consequently, to address the stringent KPIs presented in Table 1 and to allow the full implementation of some of the innovative use cases in Section 5.

### XL-MIMO

Although mMIMO considerably increases the spectral and energy efficiencies, the industry and academy have been investing in further extending the number of antennas at the BS compared to typical mMIMO systems. This new technology has been named “extremely large MIMO” [10] and is an up-and-coming technology for 6G Networks. Due to the large size of the antenna array, as illustrated in Figure 2, the XL-MIMO system can be integrated into large surfaces such as building facades, shopping malls, airports, and avenues [4,10]. In addition, they can be deployed in a centralized or distributed way across a large avenue [4]. This technology, which leads to non-stationary spatial properties, can considerably increase the throughput and spectral efficiency compared with classical mMIMO systems. Moreover, due to the more significant number of antennas at the BS, it is possible to significantly extend the coverage area and support a more substantial number of users [4].

Although the advantages previously cited, the development of XL-MIMO systems is still in its infancy, and this technology presents similar challenges as those of mMIMO. However, the hardware deficiencies, mutual coupling, channel modeling, and acquisition of CSI challenges become even more challenging due to the many elements. At the same time, research efforts are necessary as these topics represent fundamental open research problems.

## 3. Cell-Free mMIMO

A new promising approach for 6G Networks is the cell-free paradigm, in which the absence of well-defined cell boundaries reduces not only problems inherent to the handover process but also cell-edge user’s limitations and inter-cell interference [11]. The term cell-free for wireless communication networks has been defined in [39], and means that the cellular network is not divided into cells, i.e., there are no cell boundaries from a user perspective during the uplink and downlink transmissions, since all access points (AP) that affect a user will participate in the communication process. Thus, for example, when a user transmits an uplink signal, all APs that receive it, with a signal-to-interference ratio above a threshold, will collaborate in decoding the signal [11,40].

A CF-mMIMO network, as shown in Figure 3, consists of several geographically distributed APs that, together and coherently, serve a smaller number of users in the same frequency and time resource. Each AP is connected from a fronthaul link to a central processing unit (CPU), responsible for the cooperation among the APs [11]. A cell-free network can be divided into an edge and a core, like cellular networks. The AP and CPU are at the edge, while the connections between them and between the edge and the core are defined as fronthaul and backhaul links, respectively. Fronthaul links can be used to (i) share physical layer signals that will be transmitted on the downlink; (ii) forward uplink signals that have not been decoded; and (iii) share the channel state information (CSI) related to the physical channels [11]. They are also used for phase synchronization among geographically distributed APs. In a cell-free system, a CPU might not be a separate physical unit but might be seen as a logical entity. For example, CPUs can represent a set of local processors at a subset of the APs or other physical points connected through fronthaul links. In agreement with the continuous cloudification of wireless networks [41], known as cloud-radio access networks, CPU-related processing tasks can be distributed among local processors in different ways [42] and represents one of the main enablers paradigms of CF-mMIMO networks [24]. Therefore, the difference between cellular and cell-free networks is given in terms of infrastructure and signal processing, which is transparent for the user equipment (UE), i.e., the users can connect to both types of networks without making changes to their software or hardware [11].

Based on the previously described CF-mMIMO characteristics, its main advantages can be summarized as follows [43,44,45]: (i) APs and CPUs operate as a single mMIMO cell which inherits all the features of mMIMO systems [11]; (ii) it presents a distributed topology and network ultra-densification and can offer unprecedented levels of macro-diversity gain, resulting in a more reliable communication link as many APs serve each user which reduces the blocking probability. Furthermore, because the APs are closer to the users, path loss and shadowing effects are reduced, resulting in a higher channel gain [46]; (iii) the CF-mMIMO is a user-centric architecture, which is crucial to mitigate inter-cluster interference and to preserve system scalability [11]; (iv) as a result of the macro-diversity and user-centric approach, CF-mMIMO by nature guarantees a uniform quality of service for all users [46]; (v) indoor and hot-spot coverage scenarios assured seamlessly; and (vi) mobility support without overhead due to handovers [18]. This will provide stable QoS, one of the most challenging requirements envisioned for 6G scenarios in future wireless networks.

CF-mMIMO is a promising technology for future wireless networks. However, several challenges and limitations need to be considered when deploying this technology. Therefore, the main challenges of the CF-mMIMO approach are described next [11].

**Synchronization:** The CF-mMIMO network requires precise synchronization and coordination between APs, which can represent a considerable computational effort, and may require overhead signaling and significant instantaneous/statistical CSI exchanges [11];**Practical Approach:** The classical CF-mMIMO network is modeled as a single unlimited large mMIMO cell. A practical and scalable implementation must consider that the CPU and fronthaul links constitute the architectural bottleneck. Therefore, data sharing and resource allocation tasks must be performed within some APs to limit computational complexity on the CPU and signaling overhead. Moreover, fully centralized precoding and matching schemes should be avoided to overcome the need for instantaneous CSI at the CPU [11].**Hardware impairment:** The high number of multi-antenna APs in CF-mMIMO leads to significant energy consumption and hardware cost. In addition, as in mMIMO systems, a significant challenge of the CF-mMIMO is the performance degradation due to hardware deficiencies since non-ideal hardware components can add distortion noise and noise amplification at the transmitter and receiver and, consequently, distort the signal which will significantly harm the system performance [47].**Channel knowledge:** As in mMIMO systems with large antenna arrays, the perfect CSI estimation in CF-mMIMO can be impractical as the pilot transmission time may exceed the channel coherence time. However, a prerequisite to reaching all the advantages of CF-mMIMO is to have the perfect knowledge of the CSI as imperfect CSI knowledge can deteriorate the performance of the system significantly [11].

## 4. Intelligent Reflecting Surfaces

Recently proposed, IRS is a wireless communication technology that transcends the traditional concept of antenna arrays on transceivers. Being a flat matrix comprising almost passive and low-cost reflective elements with reconfigurable parameters helps interconnect the physical and digital worlds continuously and sustainably. For this reason, it is considered instrumental to 6G Networks in which the propagation environment will be transformed into a reconfigurable entity by software [48,49]. Thus, unlike conventional wireless communication systems where only the transmitter and receiver can be adjusted, IRS allows the wireless channels to be intentionally and deterministically controlled to get achievements and/or desired distributions. Furthermore, it provides new means to solve the problems of fading and interference in wireless communication channels and potentially improve signal quality at the receiver and in network capacity [12].

As illustrated in Figure 4, the IRS can be deployed on building facades, vehicles, or even drones to improve the coverage and increase the signal strength thanks to reflection. Finally, it is essential to emphasize that IRS presents itself as a conceptually attractive technology with numerous practical implementation advantages, such as [12,50,51]: (i) It is formed by reflective elements (for example, low-cost printed dipoles), which passively reflect the received signals without using RF chains for signal transmission. Therefore, they can be implemented with lower hardware costs than traditional active antenna arrays or newly proposed active surfaces. Furthermore, the exclusive use of passive elements represents a considerable reduction in energy consumption compared to classic technologies; (ii) It operates in full-duplex mode and is free of any antenna amplification noise, as well as self-interference, which represents a significant advantage when compared to traditional active relays; (iii) It features flat geometry, is lightweight, and can be easily mounted and removed from objects present in the deployment environment; (iv) It can be used as an auxiliary device in wireless networks and seamlessly integrated, providing great flexibility and compatibility with existing wireless systems.

Due to these advantages, the deployment of IRS in the future mobile networks has attracted interest from the academy and industry in the last few years. More specifically, the first studies on this subject date back to 2014. From a series of experimental implementations, in [52], the authors showed that reconfigurable reflective lens arrays could dynamically control the beam of an antenna. However, the concept similar to what we now call IRS was only mentioned in [53], its main idea consisted of electromagnetically active wallpapers with built-in processing power. The IRS controller is responsible for automatically controlling a compact integration of small antennas. Thanks to the adjustment of its elements, such as capacitance, resistance, and inductance, the IRS can change the characteristic impedance of the medium, the reflection coefficient, and, consequently, the beamforming direction. Its great advantage is the focus of energy on three-dimensional space with extreme precision, especially at high frequencies, which allows it to suppress interference. In other words, IRS improves signal coverage and increases channel capacity [54].

Nowadays, there is already much research on this topic. Most of it addresses the effects of IRS use on parameters previously developed for conventional communication systems. In addition, although the IRS advantages, this technology presents some challenges, such as [12].

**CSI Acquisition:** Estimating the CSI between the IRS and BS or between the IRS and UE is a challenge as the training overhead scales up with the number of elements at the IRS elements and can become impractical. In addition, to estimate the CSI, some IRS elements need to be equipped with RF chains which increases energy consumption which is one of the stringent requirements of the 6G Networks.**Mutual coupling:** When modeling the IRS, most works consider that the separation between the reflecting elements is large enough to keep mutual coupling at negligible levels. However, this is not entirely realistic, as the improvement in the system performance usually is obtained by increasing the number of reflecting elements at the IRS and, consequently, the distance between the elements considerably decreases, and the mutual coupling effect cannot be negligible [12].**Propagation and Channel modeling:** Under experimental conditions, the influence of the incident angle on the phase shift of the IRS elements needs to be considered, which is extremely challenging in a multi-path propagation environment [17]. In addition, it is well known that the typical time-division duplex (TDD) based wireless systems is no longer valid, which implies that we cannot consider the reciprocity between uplink and downlink during the channel estimation process [12,55]. Thus, further studies on these issues are needed.**Coupled amplitude and phase:** A practical reflection model for IRS reveals that the amplitude and phase are coupled and, consequently, cannot be adjusted independently [56]. This adds a challenge to the beamforming design, as finding an optimal balance between the signal amplitude and phase reflected by each IRS element is necessary. Then, further study on this effect and new beamforming design techniques are worth pursuing.

From the review of the main features of the mMIMO, XL-MIMO, IRS, and cell-free mMIMO technologies, we can verify that they will play an important role in the development of 6G Networks. Therefore, in the following sections, we describe the main use cases that will be supported by 6G Networks and how the MIMO technologies can help to support their requirements.

## 5. Use Cases and Key Performance Indicators (KPIs) of 6G Networks

During the definition of 5G Networks, the International Mobile Telecommunications 2020 (IMT-2020) has presented a set of use cases with very challenging requirements [57] based on demands that went beyond a higher throughput in mobile networks. The key contrasting requirements have led to three use case scenarios, known as enhanced Mobile BroadBand (eMBB) [58], Ultra-Reliable Low-Latency Communication (URLLC) [59], and massive Machine Type Communication (mMTC) [60].

The eMBB, defined by the 3rd Generation Partnership Project (3GPP) Release 15, aimed at providing data rates up to 1 Gbps per user. The URLLC, addressed by 3GPP Release 16, focused on reducing the overall latency and supporting robust communication links for industrial applications and delay-sensitive services. Finally, mMTC, which will be standardized by 3GPP Release 17, promises to support many connected devices in massive Internet of Things (IoT) applications while reducing the power consumption required for communication [58].

A critical limitation of 5G Networks that is hindering the support of all challenging applications foreseen by IMT-2020 is that 5G struggles to support, in the same network, devices with very heterogeneous requirements, e.g., high throughput and low latency [61]. As will be presented in this section, several applications for important verticals need to combine low latency, high throughput, and a massive number of connections simultaneously. Besides these limitations, new and even more challenging applications are being proposed for future mobile networks [19,23,62]. The future mobile networks will need to transcend communications and include new features, such as environmental sensing, positioning, and mapping of people and objects, imaging as a native service of the network, and the massive use of Artificial Intelligence (AI) in the entire network framework, as well as offering AI as a service for other applications [62,63]. The new use cases also will demand more friendly and intuitive ways to interact with the network that cannot be limited to gestures applied to a touch-sensitive screen. New interfaces that allow for transmitting complex commands and capturing further information, i.e., texture, pressure, temperature, motion, and biomedical signals, will be necessary to deploy several new services. The development of a new brain–computer interface is being pointed out as essential for the new use cases and applications foreseen for the 6G Networks [64]. Taking all these phases of evolution into consideration, it is possible to state that the 6G Networks will be a vehicle for the integration of the physical, biological, and virtual worlds, breaking the connectivity limitations of previous generations and changing the way that society will exploit the mobile networks in the future.

The new scope of the applications for 6G Networks is being proposed by academic research projects and associations around the world, such as [65,66,67], while, up to now, arguably the main novel applications can be categorized into four use cases. These use cases are described and the necessary KPIs are concisely presented.

### 5.1. Use Cases of 6G Networks

#### 5.1.1. Large Scale Digital Twins

The use case family denominated as Large Scale Digital Twins consists of a set of applications that demand a digital replica of a physical system, where the information flow between the physical and virtual systems happens in real time, as shown in Figure 5. With this approach, it is possible to run precise simulations and evaluate the impact of actions and policies in the virtual system before applying them in the physical one.

Large Scale Digital Twins bring several advantages for different scenarios, e.g., dynamic smart cities [68]. The expected unambiguous presence of 6G Networks and its integration with sensing, mapping imagining, and positioning will support the creation of digital twins for large complex systems, such as factories and even cities [63]. These digital twins for cities will be used to support decision-making in several areas that affect the quality of life in modern society. Some examples are mobility management (including traffic control, public transport, and routing algorithms for navigation), air and water monitoring public safety, public lights, mass communication, infrastructure management, and maintenance, among other services.

The industrial sector will also benefit from the Large Scale Digital Twins once all manufacturing processes are controlled and managed in the virtual system before the changes are applied in the factory manufacturing plant. Hence, simulations of solutions aiming for productivity improvement or new approaches to improve manufacturing performance can be tested and validated with high precision before being effectively used in the real world, allowing for a detailed evaluation of the changes in the production line. Preventive maintenance can also become more efficient since the digital twin can detect potential problems by analyzing a large set of parameters in real-time [69].

To support all requirements for this family of use cases, the 6G Networks must increase coverage and allow for data transferring among several types and classes of sensors and devices with demands that vary from low latency to high throughput. The 6G Networks also need to monitor the positioning, movements, and speeds of objects and people in a given location in real time. Otherwise, the precise mapping of the physical world in the digital counterpart will not be possible. Therefore, the Large Scale Digital Twins will push the 6G capabilities of communication, sensing, mapping, positioning, and imaging to the limit. Thus, to address these applications, it is necessary to fulfill stringent KPIs as those summarized in Table 1.

#### 5.1.2. Advanced Remote Interactions

Since the addition of bi-dimensional images in communications, resulting in audiovisual streams in teleconferences, there has been no significant improvement in the degree of immersion in remote human interactions. The Advanced Remote Interaction family of use cases aims to bring a new dimension to the way people communicate remotely, increasing immersion through extended and virtual realities and haptic communications.

The first application being foreseen for this scenario is tactile communication [62], where the goal is to make transmitting tactile information a reality, complementing the audio and video stream. In this approach, touch sensations will be sent at the same time that information about texture, temperature, and movements are received [62]. This application fundamentally changes how telecommunication networks operate because these networks were designed to transmit audio and video, while haptic communications demand instantaneous bidirectional response to provide tactile information on both sides of the connection. Low latency and high throughput must be simultaneously achieved. The ability of the 6G Networks to transmit tactile information will also trigger applications that require precise manipulation of real and virtual objects. Remote controlling heavy machinery in a hostile environment and immersive entertainment games are examples of applications that can benefit from this new capability of future mobile networks.

Considering the application for immersive events [62], the main objective is to allow people to have experiences similar to those observed in sports arenas or artistic theaters but from a remote location. Images, sounds, vibrations, and other sensory data must be acquired in the event venue and transmitted over a high-capacity network to users equipped with wearable devices (head mount display, phones, gloves, and others) that can reproduce this information. Holographic images are also being considered for this use case since spatial light modulators [70] for consumer holographic applications are being researched nowadays. In this case, the main requirement for the 6G Network is high throughput to deliver all the streams necessary to reproduce the immersive environment for the users.

In addition, the pandemic has imposed social distancing on people worldwide and it had a huge impact on society. With important commemorative events being forbidden by authorities to reduce COVID-19 transmission, it was clear that more than audio and video calls are needed to bring people together. Furthermore, with social events such as soccer matches and carnivals being canceled due to the pandemic, the demand for immersive remote events became a reality that the next generation of mobile networks must fully address.

Another important application is collaborative robotics. The most recent technological solutions in robotics have allowed the development of autonomous collaborative entities to help humans on daily tasks [71]. The main difference between conventional robots and a Cobot is that the latter can evaluate what must be done to achieve a given goal, verify what the other entities or humans are currently doing, and then decide what tasks it can perform to collaborate with the process. As depicted in Figure 6, Cobots can exchange information and establish a hierarchy to execute the tasks needed to achieve a global objective. Until recently, Cobot was employed mainly in industries and logistics, but the cost reduction in robotics enables the use of autonomous collaborative devices to perform domestic duties or to provide assistance for humans.

The presence of Cobot in homes, industries, shops, and streets is expected to increase considerably in the following years. Since these devices need to exchange a large amount of information to cooperate, the communication infrastructure must be ready to support data transfers that go way beyond the dispatch and reception of commands. It will be necessary to provide mechanisms that allow the robots to create a symbiotic relationship among themselves. This relationship can be exploited to solve complex tasks efficiently, or it can be used to support better the demands and needs of human beings in daily tasks. This level of collaboration can reduce the demand for very complex and specialized machines since Cobot can break complex tasks into several small tasks, which can be solved using a systematic approach. Furthermore, humans can be integrated into the tasks executions, and the Cobot can learn with them during this process. The flexibility of this approach can reduce resource consumption and increase the efficiency of the exploitation of autonomous devices [71]. To fully implement these new applications, 6G Networks must reach stringent requirements, which are concisely described in Table 1.

#### 5.1.3. Advanced Agribusiness

The use cases related to agribusiness are significant for many developing countries whose economy is dependent on agribusinesses, as well as for developing countries in which this sector plays an important role, such as Australia, Brazil, Chile, and the United States of America. To demonstrate the importance of this vertical, we can cite the Brazilian economy in which agribusiness represented 26.6% of the Brazilian gross domestic product (GDP) in 2020 [72]. The massive growth in the production of animal protein and grains (soybeans, corn, and others) has made Brazil one of the main players in food production worldwide. However, the increment in agribusiness production is highly correlated with the increment of the area exploited by this vertical, indicating that the productivity has only slightly increased over the years [73].

The higher demand for food and the necessity to reduce the environmental impact makes it necessary to increase agribusiness productivity to keep its role in the global food market while reducing the exploited area. Therefore, one of the main tools to achieve this goal is the informatization of the fields [74]. Unmanned aerial vehicles (UAV) with multispectral cameras can be used to capture images and detect plagues and infestations to deploy pulverization drones to apply pesticides only in the affected areas. In addition, biological sensors implanted in livestock will detect pathological agents or forbidden substances (like hormones) in the animal’s bloodstream in real time. At the same time, motion sensors can detect unusual behaviors of the animals, while AI-powered solutions can diagnose dangerous diseases based on the data collected from the cattle. These sensors can increase the traceability of the agribusiness products, reducing the impacts of eventual sanitary embargoes only to the affected region and increasing the transparency for both the final consumer and producer. Figure 7 illustrates this use case. As in the other use cases described above, to achieve the goals related to advanced agribusiness, 6G Networks must fulfill the KPIs presented in Table 1. More specifically, the 6G Networks need to provide high throughput and connectivity, allowing for a high number of devices with different data rate requirements to be deployed over a large area, while keeping an efficient connection with the BS.

#### 5.1.4. Invisible Security Zones

It is well known that many countries, mainly underdeveloped and developing ones, suffer from a high number of violent deaths. New monitoring technologies introduced by 6G Networks can help the authorities to reduce crimes and provide a new level of public safety for these populations. The capability of monitoring the environment using images and sensors introduced by the 6G Networks can provide transparent security reinforcement in public areas while also performing access control without ostentatious checkpoints. Chemical and temperature sensors, together with multispectral cameras distributed in a security area, such as an airport, as depicted in Figure 8, can provide data for AI algorithms responsible for detecting menaces and hazardous situations. In addition, chemical sensors can detect traces of forbidden substances, and the behavioral analysis of people can indicate suspicious activities. Moreover, the 6G capability to perform behavioral analysis can be used in open public spaces to detect the beginning of such activities and inform law enforcement agents to prevent crime.

To conclude, it is possible to verify that high-resolution images acquisition, diversified sensing, low latency, and high data rates are some of the contrasting requirements, summarized in Table 1, that must be supported by the 6G Networks to make the Invisible Security Zones a reality.

### 5.2. KPIs of 6G Networks

From the innovative use cases previously described, there are several wireless communications-related challenges to the design and implementation of 6G Networks. Next, we summarize the main requirements that need to be fulfilled to adequately support these use cases and their application families. More specifically, Table 1 presents the key requirements for each use case previously shown, while the description of these requirements is briefly provided next.

**Rate/User**: Typical achievable rate demanded by users.**Peak Rate**: Minimum cell peak rate.**Latency**: Maximum tolerable end-to-end delay.**Reliability**: Probability of the link being in operation and meeting the QoS requirements.**Energy Efficiency**: Energy efficiency compared to the 5G Networks, taking into account the eMBB or mMTC operating modes.**Spectral Efficiency**: Spectral efficiency compared to the 5G New Radio (5G NR) standard.**Spatial Accuracy**: Spatial accuracy of the positioning of the mobile devices.**Security:** The demand for security and protection of the transmitted data. The term “Nominal” refers to the level of security observed in 5G Networks, while “Critical” refers to a higher level of security, and may even include the physical layer security.**Privacy:** Sensitivity of the application regarding the privacy of data transmitted over the network. “Nominal” denotes the level of privacy obtained in 5G Networks while “Critical” refers to a higher level of privacy.**User Density:** Maximum number of devices per area.

The diversity of applications for 6G Networks will require a level of flexibility never seen before in any generation of wireless communication systems. In addition, these applications will demand a new set of functionalities that goes beyond the borders of communications, such as sensing the environment through the use of radio frequency (RF) signals or multispectral imaging, accurately mapping objects and people from the physical world to the virtual world, collecting real-time biological data from people and animals, and applying AI vertically across all layers of the wireless network.

From Table 1, where we map the main requirements for each family of the innovative use cases described before, we can see that conflicting goals must be met, which represents a major challenge for the different layers of access networks. Thus, 6G Networks should be a platform for integrating different technologies. The 6G Networks will permeate all these technologies, integrating satellites, UAVs, BSs, access points, and gateways of different technologies in a harmonious and synergistic environment that is effectively capable of integrating the physical, biological and virtual worlds.

Moreover, with the advancement of MIMO technologies, 6G Networks should meet many new stringent requirements and, consequently, enable some of the innovative use cases described in Section 5. The deployment of advanced mMIMO, XL-MIMO, CF-mMIMO, and IRS technologies in the architecture of 6G Networks can increase the coverage and reliability while also improving the energy and spectral efficiencies and reducing the system latency.

More specifically, mMIMO and XL-MIMO systems can achieve high data rates, spectral efficiency, energy efficiency, and reliability while enabling massive access. Then, based on the KPIs in Table 1, and as illustrated in Table 2, mMIMO and XL-MIMO systems are enablers of Large Scale Twins, Advanced Remote Interactions, and Invisible Security Zones use cases. Similar to mMIMO and XL-MIMO, CF-mMIMO can achieve high data rates, spectral efficiency, and reliability. However, differently from mMIMO and XL-MIMO, it is still unclear if CF-mMIMO can satisfy very stringent requirements in terms of energy efficiency and spatial accuracy. In addition, despite satisfying most of the Advanced Agribusiness requirements, the viability of mMIMO, XL-MIMO, and CF-mMIMO is questionable outside dense urban areas because the deployment of APs is expensive and its density must be higher than the UEs density [75]. Indeed, a key challenge is the increased deployment cost, as these technologies require a large number of antennas [7], which can be a prohibitive cost for many operators in not crowded environments.

Moreover, the use of IRS in the 6G architecture can satisfy most of the KPIs of all innovative use cases discussed in Section 5. In principle, IRS can achieve the same improvements as mMIMO, XL-MIMO, and CF-mMIMO. However, due to its flexibility, IRS can also be used as an additional tool in wireless networks and integrated into them almost transparently, providing compatibility with existing wireless systems. In addition, IRS can be combined with different technologies such as mMIMO, XL-MIMO, and CF-mMIMO, further improving their advantages and, consequently, meeting stringent requirements of several vertical applications of the use cases described in Section 5. For instance, integrating IRS with UAV communications may be vital to reaching the requirements of different agribusiness applications [76].

## 6. The Road Ahead

Although mMIMO, XL-MIMO, CF-mMIMO, and IRS have been intensively researched in the last few years, several open research topics remain to be addressed. Moreover, future communication networks are expected to be even more complex and heterogeneous, supporting increased demands of devices’ diversity. Therefore, typical management tasks may become intractable, requiring AI techniques.

Different AI techniques can be integrated into future networks to increase the flexibility of mMIMO and XL-MIMO systems [18,36], and a more accurate estimation of the CSI [77] which is primordial for the full deployment of both technologies. In addition, AI techniques can provide more efficient hardware design, reducing the distortion from the power amplifiers [36]. Moreover, signal processing schemes become even more complex as the XL-MIMO system further increases the number of antennas at the BS. Then, again, the use of AI can allow the development of novel processing schemes with acceptable complexity [4], which is an essential open research topic for the full deployment of XL-MIMO technology in 6G Networks.

In addition, the performance enhancements offered by IRS are reached by reconfiguring the wireless channels to optimize the system capacity. However, to fully achieve these improvements, considering the large size of the future wireless network and the massive deployment of IRS, new tools need to be designed [12]. Then, as for mMIMO, XL-MIMO and CF-mMIMO, AI techniques can be applied to accurately estimate the CSI and, consequently, allow the adequate design of the beamforming at the IRS for improved system performance [78]. Nevertheless, due to the massive deployment of IRS, AI can be used to deploy them optimally and appropriately cluster the users to balance the number of served users according to the power transmission constraints [79].

## 7. Conclusions

It is up to the 6G Network to provide simultaneous support for several applications that demand ultra-high data rates (in Tbps) and high coverage and availability. Novel multiple-antenna technologies, such as massive MIMO, XL-MIMO, IRS, and CF-mMIMO communications, are critical enablers of 6G Networks. Then, this paper gives an overview of each technology’s main features, advantages, and challenges. In addition, we discussed the main use cases that may be supported by 6G Networks and their specific KPIs. To finish, from the discussion presented in this paper, we can conclude that, from the deployment of multiple MIMO technologies, it is possible to reach the majority of the anticipated 6G Network requirements, allowing the support of innovative use cases that may be very relevant for future vertical applications.

## Figures and Tables

**Figure 1 sensors-23-01921-f001:**
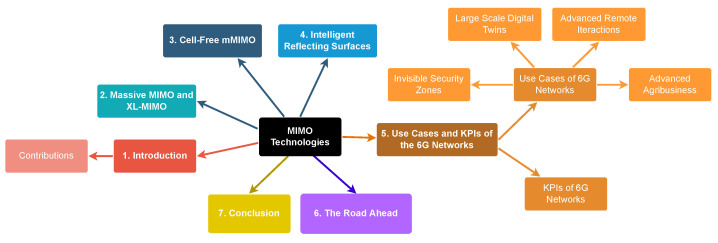
Survey organization.

**Figure 2 sensors-23-01921-f002:**
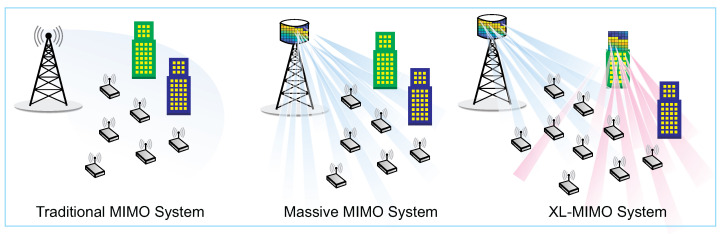
Comparison between traditional MIMO system, massive MIMO systems, and XL-MIMO systems.

**Figure 3 sensors-23-01921-f003:**
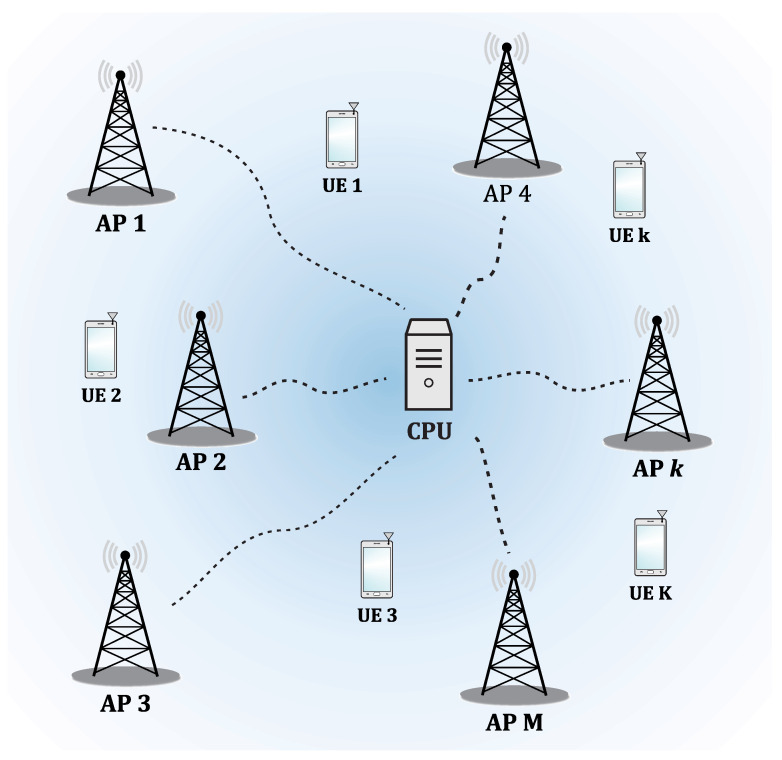
Cell-free network.

**Figure 4 sensors-23-01921-f004:**
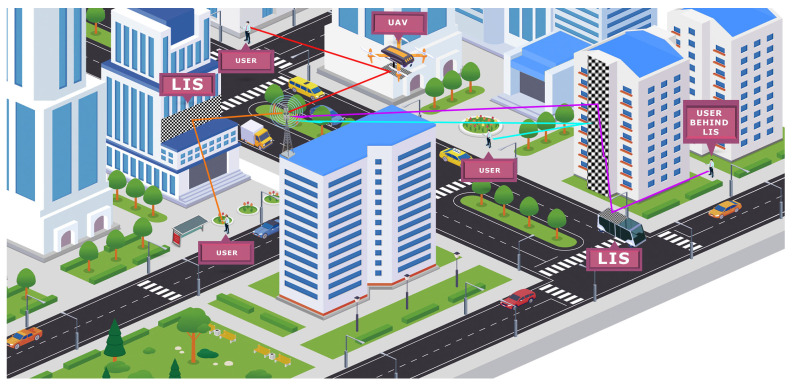
IRS applications.

**Figure 5 sensors-23-01921-f005:**
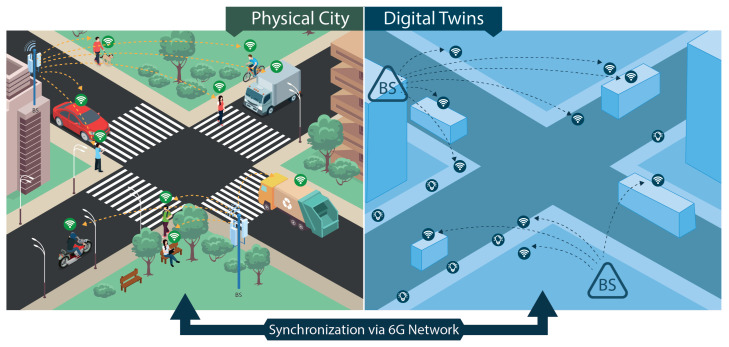
Large Scale Digital Twins use case: Physical city and its corresponding digital twin.

**Figure 6 sensors-23-01921-f006:**
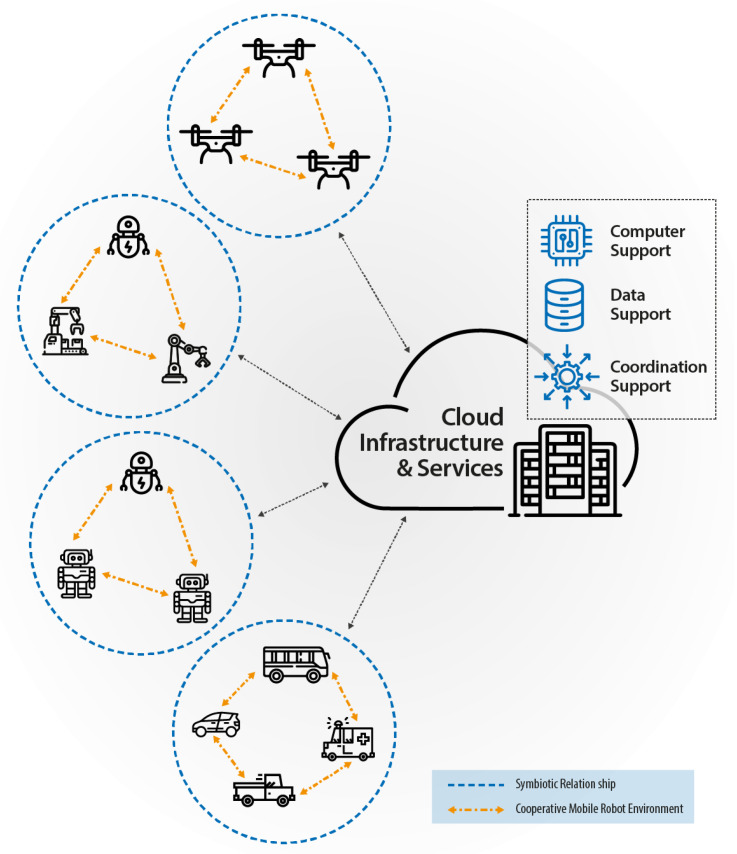
Collaborative robotics use case.

**Figure 7 sensors-23-01921-f007:**
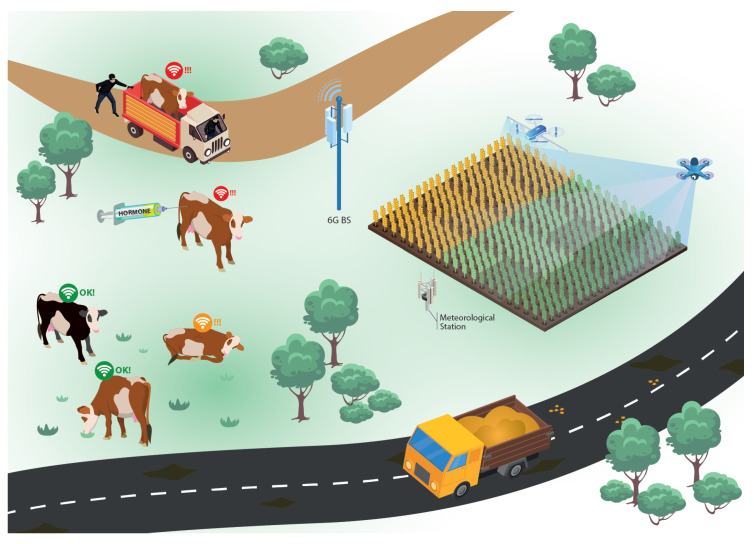
Advanced agribusiness use case: Future agriculture.

**Figure 8 sensors-23-01921-f008:**
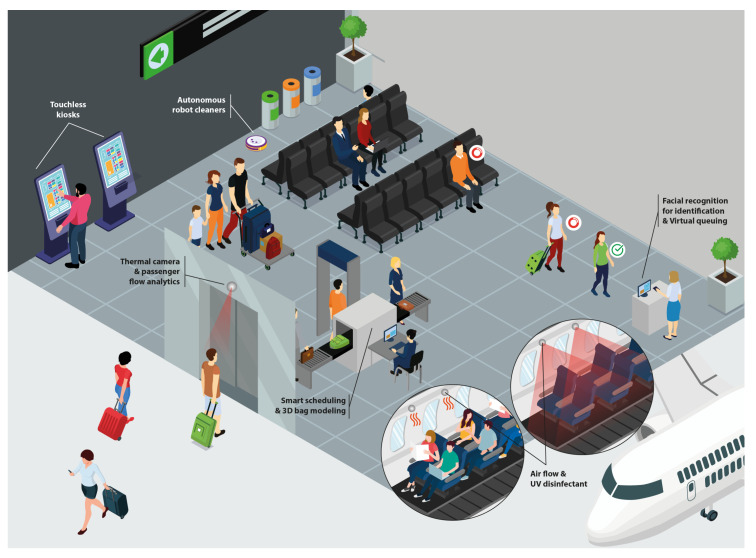
Invisible security zones use case: Invisible checkpoints for security in airports.

**Table 1 sensors-23-01921-t001:** Key performance indicators of the innovative use cases of 6G Networks.

KPI	Use Case Family			
	Large Scale Digital Twins	Advanced Remote Interactions	Advanced Agribusiness	Invisible Security Zones
Rate/User (Mbps)	$10^{-1}$ to $10^{2}$	$10^{-1}$ to $10^{3}$	10 to $10^{6}$	10 to $10^{2}$
Peak Rate (Gbps)	>1	>100	>$10^{4}$	>100
Latency (ms)	<1	<1	<1	<20
Reliability	$1-10^{-5}$	$1-10^{-9}$	$1-10^{-9}$	$1-10^{-7}$
Energy Efficiency	10x mMTC	10x eMBB	10x eMBB	10x eMBB
Spectral efficiency (in relation to 5G NR)	1x	10x	10x	10x
Spatial Accuracy (cm)	<10	<1	<1	<10
Security & Privacy	Nominal	Critical	Critical	Critical
User Density (1/${\mathrm{km}}^{2}$)	100	$10^{4}$	$10^{4}$	10

**Table 2 sensors-23-01921-t002:** Use cases in Section 5 that can be supported by the MIMO technologies discussed in this work.

Use Cases	Massive MIMO	CF-mMIMO	XL MIMO	IRS
Large Scale Digital Twins	✓	✓	✓	✓
Advanced Remote Interactions	✓	✓	✓	✓
Advanced Agribusiness	✗	✗	✗	✓
Invisible Security Zones	✓	✓	✓	✓

## Data Availability

Not applicable.

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
