# Peer review of "Emerging MIMO Technologies for 6G Networks"

_sensors, 2023, doi:10.3390/s23041921_

Round 1

Reviewer 1 Report

The manuscript contains in-depth and comprehensive survey of the MIMO transmission techniques for upcoming 6G mobile networks and related future research issues. One drawback of the paper organization noticed by the reviewer is that the Use Cases in Section 2 and MIMO techniques introduced in Sections 3 to 5 are not very well inter-related.

I suggest that the authors describe the techniques in Sections 3 to 5 first, then explain how each of these techniques can be applied to future use cases with more details on requirements and solution strategies. In this suggested organization, Sections 2 and 6 in the current manuscript can be combined and placed after introducing the mMIMO, XL-MIMO, CF-mMIMO, and IRS. A table-type of explanation on which MIMO techniques would be particularly beneficial to each of use cases would be helpful to the readers to understand the advantage of the enhanced MIMO techniques. 

Author Response

Dear Reviewer,

We thank you for your generosity and precious time, and for giving us the opportunity to highlight and improve the contributions of our work. All the changes in the revised manuscript are marked in blue to facilitate the review process.

Please find attached, a document with each of the Reviewers’ comments.

Best Regards,

Victoria Souto and co-authors.

Reviewer 2 Report

This survey gave a concise review of the main emerging MIMO technologies for 6G Networks such as massive MIMO (mMIMO), extremely-large MIMO (XL-MIMO), Intelligent Reflecting Surfaces (IRS), and Cell-Free mMIMO (CF-mMIMO). A survey article should be more comprehensive and provide insight comparisons which not available here. After carefully reading, seen that this article did not add a new to the literature.

Author Response

(The authors gave the same response as above.)

Round 2

Reviewer 2 Report

The revised article is improved and clear and can be accepted to publish